# OpenReview forum: "Interpretability of LLM Deception: Universal Motif"
_ICLR.cc/2025/Conference — Submitted to ICLR 2025_

### Official Review · Reviewer_8jsY · 2024-11-02

**Soundness:** 3
**Presentation:** 2
**Contribution:** 3
**Rating:** 6
**Confidence:** 3

**Summary:**

This paper investigates several large language models (LLMs) to understand when and why these models may exhibit deceptive behavior. The authors first find that the tendency to lie increases with model size. They then explore how latent representations associated with lying evolve through three iterative refinement stages, concluding that smaller models lack the ability to lie as they cannot rotate truth directions during the third stage. The study further examines if this third stage is causally linked to lying, with findings that suggest a causal relationship.

**Strengths:**

1. The study examines a diverse range of language models across various sizes and model families.
2. The research findings are intriguing, particularly the observation of truth direction rotation in the third refinement stage for models capable of lying.
3. The verification of findings, especially the conclusion that the third stage is causally linked to lying, appears robust and well-supported.

**Weaknesses:**

1. The protocol used to instruct models to lie intentionally is overly simplistic. As a result, the conclusion that lying scales with model size may be somewhat limited. It would be valuable to explore whether smaller models could also exhibit deceptive behaviors if prompted with more sophisticated, carefully engineered instructions.

2. While it is interesting to identifying that the third refinement stage in the rotation of truth direction is causally linked to a model's ability to deceive, it would add depth to the study to investigate why smaller models fail to achieve this directional rotation.

3. The paper's readability, particularly in Section 4.4 and Figure 5, could be significantly improved. It is challenging to understand how the results presented in Figure 5, along with the text in the second paragraph of Section 4.4, effectively demonstrate the causal relationship between the third refinement stage and lying behavior.

**Questions:**

N/A

---

> ### Author Response · Authors · 2024-11-21
>
> We sincerely thank the reviewer for taking the time to thoroughly review our paper and for providing thoughtful and insightful feedback. We greatly appreciate the reviewer’s recognition of our discovery of the truth direction rotation as "intriguing" and their acknowledgment that our experimental verifications of their causal role are "robust and well-supported."
>
> In response to the reviewer’s comments, we have incorporated new experiments into the manuscript to address their concerns. We believe these additions further strengthen the paper and enhance its potential for acceptance. Below, we provide a detailed, point-by-point response to each of their comments:
>
> ---
> ### 1. Prompting protocol.
> Thanks very much for pointing out that our original prompt design might seem too “simplistic”. In response to the reviewers’ comment, we have now added additional results in Appendix B. We agree that it is immensely valuable to explore whether smaller models could exhibit deceptive behaviors if prompted with more sophisticated, carefully engineered instructions. Following the reviewer’s insightful suggestion, we have added new experiments to rule out the possibility that more sophisticated prompts can induce small models to lie.
>
> We evaluated 100 prompts generated by GPT-4o, along with six human-engineered prompts proposed by Guo et al. (2023). This resulted in a diverse set of prompts employing a wide range of prompting strategies, including but not limited to:
>
> - Persona modulation (e.g., role-playing as a storyteller, Sci-Fi writer, or comedian).
> - Contextual framing (e.g., hypothetical scenarios, alternative realities, or playful settings).
>
> Our original conclusion that smaller models cannot lie (comparisons are made within the same model family and generation) remains valid.
>
> We'd like to highlight that while some prompts reduced accuracy to chance levels, small models still failed to demonstrate lying behavior. For example, a specific GPT-4o-generated prompt yielded an accuracy of chance level for the Yi-6b model, suggesting at first glance that the small model might be capable of lying. However, further analysis of the confusion matrix showed that the model primarily outputs either "True" or predominately "False", regardless of the actual labels of the statements. This kind of behavior does not qualify as lying.  All 106 tested prompts are documented in Table 3, Appendix B to ensure transparency and reproducibility.
>
> Finally, we would like to highlight that the simplicity and generalizability of our protocol are strengths rather than limitations. For comparison, consider the widely adopted chain-of-thought (CoT) prompting technique. While the CoT prompt ("Let's think step by step") may appear simplistic, its power lies in its ability to reliably elicit chain-of-thought reasoning across a broad spectrum of models.  This approach not only provides a reliable framework for eliciting lying behavior in a large set of models but also establishes a controlled environment for dissecting the causal mechanisms underlying such behaviors.
>
>
> ---
>
> ### 2. Why small model cannot lie
>
> We greatly appreciate the positive feedback on our discovery that the third stage plays a causal role in lying. Addressing the "why" question will be a central focus of our future work, as it often represents one of the most challenging aspects of deep learning research.
>
> Nevertheless, we believe the findings presented in this paper mark an important first step toward understanding how LLMs lie. We are very excited about our novel findings and by sharing these insights with the broader research community, we hope to inspire further investigations and catalyze research to understand "why" small models cannot lie.
>
> ---
>
> ### 3. Writing
>
> Thanks for for your suggestion! We will make the writing in Section 4.4 and Figure 5 clear for those who are not familiar with patching and activation steering in the camera-ready version of the paper.
>
> ---
> Lastly, we want to thank the reviewer once again for their thorough and thoughtful feedback. We kindly ask them to reconsider their score in light of the new additions, particularly the inclusion of a diverse set of prompts, which we believe have significantly strengthened the validity of our claim.

---

> > ### Author Response · Authors · 2024-11-28
> >
> > Dear Reviewer  8jsY,
> >
> > We wanted to say in advance our heartfelt thank you's for the time and effort you've put into both the reviews that have passed as well as the upcoming discussion period. We know that you all have your own papers that you have to deal with during this busy time, and sincerely appreciate the time you've taken to spend on ours.
> >
> > We are so excited about this paper and its findings and have very much enjoyed this constructive scientific dialogue with you! Please do not hesitate to ask us any questions big or small, and we are happy to provide any further clarifications.

---

### Official Review · Reviewer_FK9k · 2024-11-03

**Soundness:** 1
**Presentation:** 1
**Contribution:** 1
**Rating:** 1
**Confidence:** 5

**Summary:**

First conduct a simple prompt engineering to induce LLMs from four families (Qwen, Yi, Llama and Gemma) with different sizes to knowingly lie.
Then use the simple key word matching ("true"/"false") proportion as the evaluate metric for LLM deception.
Using the well-kown interpretability tool, i.e., activation steering, to study the key behind deception behavior and try to reduce deception.
Name 3 stages and some directions, give the finding that if a LLM complete the third stage by rotating the truth direction, it can perform deception knowingly.

**Strengths:**

Study an important problem of safety.
Do some good visualization.

**Weaknesses:**

1. The paper writing is not clear. Introduction part is too short to introduce the motivation and proposed method. Arrange of this paper is also bad. Figure 1 is far away from its corresponding explanations. There are also some typos resulting in difficulty in reading.
2. The authors do simple prompt engineering to induce lies from various models. They reach the susceptable conclusion: small models ''cannot'' lie. They cannot conclude like this unless exhausitive induction. I highly suspect this conclusion is incorrect if you try another prompt engineering approach.
3. LLMs' responses can vary a lot with multiple runs even with the same prompt. Results in Figure 2 with the simple metric ''accuracy'' are not reliable in this circumstance. And the authors say nothing about specific temperatures for these LLMs, which further makes the results untrustworthy.
4. With little innovation, this paper just uses a famous interpretability tool to study the deception scenario, presenting some findings seems not robust or reliable.

**Questions:**

1. Can you explain why the four subfigures in Figure 4 part A are totally identical?

---

> ### Author Response · Authors · 2024-11-21
>
> We appreciate the reviewer’s comments and feedback and would like to offer our heartfelt thanks for the time they spent on our paper.  We would like to point out, however, that we respectfully disagree with their assessment of our paper.  The Reviewer’s main point, as far as we are able to see, is that our results are 'not trustworthy'. We kindly invite the reviewer to provide more specific, evidence-based arguments, especially in light of our additional analysis and clarifications , as this is a grave matter. This is especially important to us as scientists as we have provided all code and analysis for full replicability. In addition, we would very much appreciate it if the kind Reviewer could clarify why employing a "popular interpretability tool" renders our findings “not robust or reliable”. Consistent with our own high scientific standards, we have reported the full statistics on all our experimental results from 24 different LLMs.
>
> Our comprehensive response is below. We sincerely hope that the Reviewer may enjoy the novel and interesting nature of our scientific results and appreciate their robustness and thoroughness. We remain enthusiastic and stand ready to address any remaining concerns the Reviewer may bring to us.
>
>
> ## Addressing Weakness
> ### 1. Regarding Clarity
> >"The paper writing is not clear. Arrange of this paper is also bad. Figure 1 is far away from its corresponding explanations."
>
> We thank the reviewer for pointing out the problem in the clarity of our writing. Our goal is to ensure the clarity and accessibility of our work, so we would endeavor to improve the clarity and organization of our paper in every possible way.
>
> Writing an easy-to-understand introduction is indeed important, especially if we want to reach a wider audience who might not be familiar with the field. We will carefully revise the introduction in the camera-ready version to make it more accessible and easier to understand for readers who may be unfamiliar with the field and its methods.
> We kindly ask the reviewer to clarify which parts of the writing they found unclear. We were heartened to find that both Reviewer nvCr and Reviewer mAEk have noted that our writing and presentation are clear. In particular, Reviewer nvCr explicitly highlighted clarity as one of the strengths of our manuscript’, stating:
>
> "The figures are exceptionally clear at illustrating the rotation effect and the distinctions between the phases. The math explaining the computation of the activation steering vector is easy to follow."
>
> We will make sure that Figure 1 will be close to its corresponding explanations.
>
> >"Introduction part is too short to introduce the motivation and proposed method."
>
> We are happy to revise the introduction in the camera-ready version to make it more accessible and easier to understand for readers who may be unfamiliar with the field and its methods.
>
> > "There are also some typos resulting in difficulty in reading."
>
> We acknowledge that there may be one or two minor typos in original manuscript and we apologize for it. We will meticulously check and double check to ensure that the camera-ready version of the paper is typo-free.  However, we hope that the minor mistakes and typos in the manuscript did not cause "difficulty in reading". If they did cause major understanding problem, we kindly ask the reviewer to provide examples of sentences or sections that were challenging to understand or difficult to follow. We will make sure to improve any specific problems that the reviewer further point out. This kind of constructive feedback would help us make targeted improvements and happily improve the quality of the paper even further.
>
>
> ---

---

> ### Author Response · Authors · 2024-11-21
>
> ### 2. Regarding Prompting
>
> >"They reach the suspectable conclusion: small models ''cannot'' lie. They cannot conclude like this unless exhausitive induction. I highly suspect this conclusion is incorrect if you try another prompt engineering approach."
>
> We thank the reviewer for bringing up this important potential problem regarding prompt engineering. Thanks to the reviewer’s comment, we have conducted careful investigation to make sure that our original conclusion is correct. We would like to express our gratitude to the reviewer, as it  gave us a chance to thoroughly exclude alternative possibilities and address potential caveats.
>
> To rule out the possibility that sophisticated prompts can induce small models to lie, we evaluated 100 prompts generated by GPT-4o, along with six human-engineered prompts proposed by Guo et al. (2023). This resulted in a diverse set of prompts employing a wide range of prompting strategies, including but not limited to:
>
> - Persona modulation (e.g., role-playing as a storyteller, Sci-Fi writer, or comedian).
> - Contextual framing (e.g., hypothetical scenarios, alternative realities, or playful settings).
>
> Our original conclusion that smaller models cannot lie (when comparisons are made within the same model family and generation) remains valid. A detailed examination of our results reveals that, while some prompts reduced accuracy to chance levels, small models still failed to demonstrate lying behavior. For example, a specific GPT-4o-generated prompt yielded an accuracy around 50% for the Yi-6b model, suggesting at first glance that the small model might be capable of lying. However, further analysis of the confusion matrix showed that the model primarily outputs "True" regardless of the actual labels of the statements, indicating that the behavior does not qualify as lying. These findings have been added to Appendix B, and all 106 tested prompts are documented in Table 3 to ensure transparency and reproducibility.
>
> Once again, even though our original conclusion that small models could not lie still holds, we found this exercise very useful, as it actually helped to strengthen our claim. We greatly appreciate the reviewer’s input as it stimulates us to add this important investigation to our paper. We hope the reviewer will reassess their original suspicion in light of our additional results.
>
> ### 3. Regarding stochasticity and temperature
>
> We thank the reviewer for their comment. The reviewer is correct that model responses can indeed be stochastic, varying from run to run. This variability underscores the importance of statistical measures, such as the mean and variance, calculated over large datasets. This is precisely what we did in our original manuscript. In addition, we also carefully examined the confusion matrix of the generated result to make sure the models that appear to be lying were not simply uniformly output "true" or "false" to all questions (Appendix E).
>
>  We do not think that stochasticity in model behavior can explain away our findings, as our analysis is based on a large dataset. Specifically, we have tested our hypothesis on 4,629 statements spanning diverse categories, including:
>
> - 1,458 questions about cities
>
> - 777 questions about companies
>
> - 472 statements about scientific facts
>
> - 594 statements regarding inventions
>
> - 692 questions about animals
>
> - 636 questions about elements.
>
> These details, along with example prompts, are provided in Table 2, Appendix A of the updated manuscript.
>
> In our experiments, temperature =1.0 for model generation. We have provided the hyperparameters in Table 4, Appendix D.
>
> We are grateful that the reviewer carefully thought about important details like  the temperature. Thanks to the reviewer’s comment, we systematically varied the temperature parameter and found that our conclusions remained consistent across all settings. For example, in Figure 7 of Appendix D, we demonstrate that Yi-6B consistently fails to lie across temperature settings ranging from 0 to 2 (with a step size of 0.2).
>
> - We emphasize that our conclusion is robust:
> 	- Consistent across 24 models of varying sizes and model families,
> 	- Robustly tested on a big dataset (with 4629 questions from 6 diverse categories),
> 	- Carefully tested with various temperature settings for generation.
>
> In light of the expanded dataset and systematic variation of temperature, which are new experiments we have conducted inspired by the Reviewer’s comments , we kindly hope the Reviewer may reconsider their original assessment of our paper.
>
> ---

---

> > ### Comment · Reviewer_FK9k · 2024-11-25
> >
> > Since when a dataset with only 4629 questions became a big dataset?

---

> ### Author Response · Authors · 2024-11-21
>
> ### 4. Regarding innovation and robustness
>
>
> > "With little innotation, this paper just uses a famous interpretability tool to study the deception scenario, presenting some findings seems not robust or reliable."
>
> We thank the reviewer for their comment. We have worked hard to thoroughly dissect this comment point by points as follows:
>
> (1). Reviewer FK9k seems to have missed the core innovation of our work: our originality lies in uncovering the three-stage process of lying and establishing its causal relationship with deceptive behavior. Characterizing our study as simply applying “a famous interpretability tool to study the deception scenario” is a significant misrepresentation of our findings. More importantly, it runs counter to the rich tradition of empirical science, which relies on using established tools to explore new and meaningful scenarios, and make insightful observations. This is exactly what we set out to do - using “famous interpretability tools" to study the “deception scenario”, discovering the three-stage process of lying and establishing its causal relationship with deceptive behavior. We believe that the true measure of innovation lies in what has been newly discovered and we are deeply excited to share our intriguing and impactful findings with the scientific community.
>
> We would like to humbly point to our other esteemed Reviewers who have enjoyed our empirical results as both intriguing and novel:
>
> - Reviewer nvCR highlighted that:
> 	- "The model finds a novel rotation effect in latent space of models when prompted on a honesty and lying dataset. Though related work has engaged with Deception in various ways, this type of interpretability using activation steering has seen little precedent in deception research..."
> 	- "This work presents an interesting starting point for lots of future work investigating the nature of deception in LLMs..."
> 	- "This work has found a highly unique pattern which may allow for safety intervention based on activations for models in inference."
>
> - Reviewer mAEk remarked:
> 	- "The findings on the three stages of activations across layers are original."
>
> - Reviewer "8jsY" commented:
> 	- "The research findings are intriguing, particularly the observation of truth direction rotation in the third refinement stage for models capable of lying.
> 	-  "The verification of findings, especially the conclusion that the third stage is causally linked to lying, appears robust and well-supported."
>
>
>
> ---
> > "this paper just uses a famous interpretability tool to study the deception scenario, presenting some findings seems not robust or reliable."
>
> (2). We respectfully disagree with the assertion that employing a "popular interpretability tool" renders our findings “not robust or reliable.” Robustness is determined by the soundness of our methodology and the consistency of our results, not by the novelty of the tools we use. In fact, leveraging a well-established method enhances the credibility of our conclusions rather than weakens them, as it ensures that our findings are grounded in reliable and widely accepted techniques.
>
> ---
> > “Using the well-kown interpretability tool, i.e., activation steering, to study the key behind deception behavior… With little innotation, this paper just uses a famous interpretability tool to study the deception scenario…”
>
> (3). Lastly, the comment that we _“**just** use activation steering”_ suggest that the reviewer have overlooked our patching experiments all together, which constitute a significant part of our work. We devoted an entire section and dedicated a whole figure to these experiments (Fig. 5, and Section 4.4), which go beyond activation steering to offer additional evidence supporting the causal role of the stage three layers in lying behavior.
>
> ---
>
> We hope this clarifies any misunderstandings and highlights the novelty and rigor of our approach. We stand ready to address any additional concerns the reviewer may have, and would greatly appreciate more specific and evidence-based feedback to ensure a constructive dialogue. In light of the additional experiments and clarifications, we sincerely hope that the Reviewer may enjoy the novel and interesting nature of our scientific results and appreciate their robustness and thoroughness.

---

> ### Author Response · Authors · 2024-11-21
>
> ### Addressing Questions
>
> > 1. Can you explain why the four subfigures in Figure 4 part A are totally identical?
>
> We invite the reviewer to take a closer look, as the panels may appear similar but are certainly not identical.
>
> - Firstly, the axes differ: Panel A presents results from LLaMA-3-8B, which has 32 layers, while Panel B shows results from Gemma-2-9B, which has 42 layers.
>
> -  Additionally, the average logit difference around the last token is distinct between the two panels (a closer examination will reveal these differences, even though they may look similar at first glance).
>
> We want to highlight that the similarity presented here underscores the robust consistency of our findings across different models of families.
>
>
> ---
> Altogether, we are grateful for the Reviewer’s comments, and their resulting comments have inspired us to diagnose possible sources of clarification and have resulted in the sharpening of the manuscript's clarity. Moreover, we are happy to report that we have completed new experiments and new analyses to address Reviewer’s comments, which have, altogether, created a manuscript even stronger than before, which we thank the Reviewer for.
> We hope the Reviewer will recommend acceptance for this paper due to its novel insights and very robust (consistent across 24 models), replicable  (with publicly provided codebase) results, relevant for an extremely important topic of our time (AI safety in LLMs). We are very excited to present our results to the scientific community, and we believe this paper will have positive impact on deeply understand how deception emerge in LLMs. We stand ready to address any remaining questions.

---

> ### Comment · Reviewer_FK9k · 2024-11-25
>
> Have you proposed any new interpretability methods? If not, how can you claim novelty once again?

---

> ### Author Response · Authors · 2024-11-25
>
> > Since when a dataset with only 4629 questions became a big dataset?
>
>
> Below, we highlight several published papers—some of them highly cited in the field—and the sizes of their respective datasets:
>
> A more meaningful comparison would be with dataset sizes commonly used in interpretability or alignment studies. Below, we highlight several published papers—some of them highly cited in the field—and the sizes of their datasets:
>
>
> -  Arditi et al. (2024): Constructed their training set by randomly sampling 128 harmful instructions from ADVBENCH, MALICIOUSINSTRUCT, and HARMBENCH to create steering vectors for refusal.
>
> -  Chao et al. (2024): Developed JAILBREAKBENCH, a dataset containing 100 harmful instructions.
>
> -  TruthfulQA: This benchmark, cited over 1,267 times, evaluates language models on their ability to generate truthful answers in a zero-shot setting. It includes 817 questions spanning 38 categories.
>
> Along the same vein as these wonderful and insightful papers from recent years, we believe we may call our dataset with 4,629 questions as relatively large within the context of interpretability and alignment research.

---

> ### Author Response · Authors · 2024-11-25
>
> > "Have you proposed any new interpretability methods? If not, how can you claim novelty once again?"
>
> We thank the Reviewer for this point. We would like to reiterate once more that our novelty lies not in proposing any new interpretability techniques, but in leveraging existing tools to uncover a robust, universal pattern and shared mechanism underlying lying in various LLMs.
>
> As pointed out by other Reviewers, understanding how lying and deception emerge in LLMs is a crucial question. Our discovery of the three-stage process of lying, along with its causal relationship to deceptive behavior, represents a novel and significant contribution to the field with practical impact in an area (LLM safety) that is very relevant in our times.
>
> On a serious note, we would like to point out that dismissal of a paper that does not “propose new ... methods” as lacking “novelty” is a dangerous precedent for the empirical sciences that we firmly believe should not happen.
>
> We sincerely hope the Reviewer may judge our paper based on objective scientific standards. We stand ready to answer any additional questions the Reviewer may have.

---

### Official Review · Reviewer_mAEk · 2024-11-04

**Soundness:** 3
**Presentation:** 1
**Contribution:** 3
**Rating:** 5
**Confidence:** 4

**Summary:**

The paper studies the internal representation produced by LLMs prompted to lie or respond truthfully to binary questions. In particular, they analyse the activations at all layers of the model and plot the first 2 principal components (PCs) of the activations. The resulting scatterplot can be grouped in three stages: first, the PCs of the activations corresponding to truthful and untruthful scenarios separate; then, within each of the clusters obtained in the first stage, the activations corresponding to true and false answers separate, with the vector going from the centroid of the true to the false cluster ("truth direction") being roughly parallel across truthful/untruthful scenarios; finally, the activations for the true and false answers in the untruthful scenarios get swapped, thus leading to antiparallel truth directions. These three stages occur in all models capable of lying in the ones they tested. Next, they perform some experiments to probe the causal nature of the rotation: first, they perform "patching", where the activations corresponding to the lying scenarios are patched onto the truthful scenario to see if the model ends up lying or not; they find that a small set of token positions and heads lead to changing model behaviour. Finally, they perform model steering, namely, adding a constant steering vector obtained as the average difference of the activations in the truthful and lying scenarios; they find that only steering the layers from the third stage reduces lying. Therefore, they conclude that these layers are causally responsible for the lying behaviour.

**Strengths:**

## originality
- the findings on the three stages of activations across layers are original.

## quality
- The experiments are thoroughly conducted and analyzed.

## clarity
- The various figures present the main findings very clearly and succinctly.

## significance
- Understanding how lying and deception emerge in LLMs is an important question; moreover, the paper also suggests that activation steering could neutralise (or at least reduce the prevalence) of deception.

**Weaknesses:**

- The related works are missing reference to https://arxiv.org/abs/2407.12831, which shares several similarities with the paper (in particular, they also identify a 2-dimensional space which represents lying/truthful state and true/false answer.

- Some parts of the paper are written in a too informal manner for a published work, and seem rather to have been obtained from internal research notes. In particular, I refer to Sections 3.4, 3.5, 3.6, 3.7, 4.2, 4.4.

- Other minor presentation details: Section 3 could introduce what the various subsections should discuss; Fig6B has a "Loading Mathjax" box which should be removed.

- The paper claims that it "introduces a simple yet general protocol to induce large conversational models to knowingly lie"; however, from the description in 3.2, this protocol seems extremely simplistic (two prompt templates fixed across LLMs), not advancing in any way with respect to previous settings, such as Pacchiardi et al. 2023. As such, I don't think this should be presented as a main contribution, nor as "careful prompting design", as it is described in Sec 3.3. Or, does the description of Sec 3.2 overlook important details (see question below)?

- The paper claims that the patterns they find are "universal". While interesting that they are coherent across the considered LLMs, the setup they consider is still quite narrow (a single prompting setting, and only ~100 binary questions linked to scientific facts). As such, I believe the use of the term "universal" is unsuitable, particularly in Secs 2 and 5.

- Relatedly, the fact that a fairly narrow set of examples is considered makes me wonder how general the patterns of activations (shown in Fig 3) are; in particular, it may as well be that the activations are different for a different set of samples. Or, it may be that the overall patterns are preserved, but that the different clusters are less distinct due to larger variety in the prompts.


- More experiments could be done to complement the ones provided and provide additional evidence of the causal nature of the 3-stage patterns identified. For instance:
	- To complement the study in Sec 4.5, the authors could try steering honest models to lie.
	- Also, steering using the truth direction rather than the honest direction could be done

**Questions:**

Sec 3.2:
- is the "protocol" only composed of those two fixed prompt templates, or are other templates used (for instance, for different models)?

Sec 3.3:
- Is the "match" with the ground-truth label done as a simple substring check? If that is the case, "The answer is not *true" would be incorrectly marked.

Sec 3.4:
- Truth direction: what do "true" and "false" refer to? Is it the answer that the LLM produces, or the ground truth?
- That is an arithmetic mean, not a geometric one.

Sec 3.5.1:
- It is not clear from the formulas there that the prompts are contrastive; in particular, the definitions in Eq 4 do not make clear that the same number of prompts are used.

Fig 1:
- The lie example in Fig 1 is not very "convincing", but it is rather as if the model was joking. Are all generated lies across models of this form? If that is the case, then maybe the authors could try methods that generate more convincing lies. If that is not the case, then it would be interesting to investigate if there is any distinction in the activations between convincing and non-convincing lies.

Sec 4.3:
- have the authors got any interpretation for why the "truth direction" influences the ability to lie?

Sec 4.4:
- is the patching experiment done on all models? Or only on those that are capable or incapable of lying?
- if the activations of a complete layer are replaced, does that mean that all downstream activations are equivalent to those that the model would produce if the truthful example was presented to the model?

---

> ### Author Response · Authors · 2024-11-21
>
> We appreciate the thoughtful and constructive feedback from the reviewer. We appreciate the positive comments and the resonance you have with us on the novelty of our findings, the robustness of our results, and the practicality of contrastive activation steering for mitigating lying behavior in LLMs.
>
>
> In response to the reviewer's feedback, we have made several improvements to the manuscript. These include the addition of new figures and sub-panels in the Appendix, as well as an expanded dataset, all of which are incorporated into the revised version of the paper.  Below are our point-to-point response to their comments:
>
> ---
>
> ## Addressing weaknesses:
>
> ### 1. Regarding Missing Reference:
>
> We thank the reviewer for bringing this additional literature to our attention. We have now included this work in the reference list of the revised manuscript. We are pleased to note that this reference aligns with our findings and those of others, highlighting the remarkable universality of two-dimensional representations of truth. However, we would like to emphasize several key innovation of our paper:
>
> - 1. **Novel Observation**: Our discovery of the third stage of change—specifically, the rotation of the "truth direction" within a narrow subset of middle-to-late layers—is novel and has not been reported in prior works.
> - 2. **Causal Analysis**: We further confirmed the functional significance of these third-stage layers through causal analysis, employing techniques such as patching and contrastive activation steering. These methods demonstrate the critical role these layers play in enabling lying behavior.
> - 3. **Extensive Model Testing**: Our conclusions are based on extensive testing across more than 24 models from four different model families (we added 4 more models comparing to the original submission), with parameter sizes ranging from 1B to 72B. The consistent patterns observed among all models capable of lying underscore the robustness of our findings.
>
> ---
>
> ### 2. Regarding writing and presentation
>
> - In the camera-ready version, we will include an introduction outlining the topics covered in the various subsections of the paper, as suggested by the reviewer.
> - We will revise the writing in Sections 3.4, 3.5, 3.6, 3.7, 4.2, and 4.4 to improve clarity in the camera-ready version of the paper,  with a particular focus on improving accessibility for readers unfamiliar with activation steering and patching. We have made a figure (Fig. 12) in appendix I to explain the patching procedure in detail.
> - Thank you for pointing out this minor issue. We have resolved the problem with the "Loading MathJax" box in Figure 6B, and it has now been removed.

---

> ### Author Response · Authors · 2024-11-21
>
> ### 3. Regarding prompting design
>
> Thanks very much for pointing out that our original prompt design might seem too “simplistic”. In response to the reviewers’ comment, we have now added additional results in Appendix B.
>
> We wanted to clarify that our carefully designed prompts were instrumental in enabling the study of lying behavior across a broad range of models with varying sizes and families.
>
> While Pacchiardi et al. (2023) focused on prompting a single large, proprietary model—GPT-3.5—to lie, our work aimed to establish a **general** protocol capable of _consistently eliciting lying across models of diverse architectures and parameter counts_. As highlighted by Guo et al. (2023), inducing smaller open-source models to lie presents a non-trivial challenge.
>
> To the best of our knowledge, our study is the only work in the deception literature that applies a consistent experimental setup to induce lying across a wide range of models of different sizes and families, thereby demonstrating the robustness of our observations.
>
> Poorly designed prompts can lead models to appear as though they exhibit lying behavior when evaluated solely on accuracy metrics. However, closer analysis often reveals that small models may simply output "True" or "False" indiscriminately, leading to low accuracy without genuinely engaging in lying behavior.
>
> To ensure the robustness of the prompt design, we evaluated 100 prompts generated by GPT-4o, alongside 6 human-engineered prompts proposed by Guo et al. (2023), resulting in a diverse set of prompts with diverse prompting strategies, including but not limited to:
>
> - Persona modulation (e.g., role-playing as a storyteller, Sci-Fi writer, or comedian).
> - Contextual framing (e.g., hypothetical scenarios, alternative realities, or playful settings).
>
> All 106 prompts tested are documented in Table 3 of Appendix B for transparency and reproducibility. We will highlight this point in the camera-ready version of the paper.
>
> Finally, we would like to highlight that the simplicity of our protocol is a strength, not a limitation. As an analogy, consider the widely recognized chain-of-thought (CoT) prompting technique. The CoT prompt (e.g., "Let's think step by step") may appear simplistic, yet its power and elegance lie in its ability to reliably induce chain-of-thought reasoning across a wide variety of models. Similarly, our simple yet general prompt reliably elicits lying behavior across models of vastly different sizes.
>
> Moreover, this simple and generalizable setup offers a controlled environment for investigating the underlying mechanisms and identifying the causal components that contribute to lying behavior. This approach enables deeper insights while ensuring reproducibility and broad applicability across diverse model architectures.
>
> ---
>
> ### 4. Regarding "Universal"
>
> We are humbly grateful for the Reviewer’s careful reading of the paper and the thoughtful suggestion to test a diverse set of questions beyond scientific facts.
>
> -  Thanks to the reviewer's suggestion, we have expanded our dataset to include statements of **six diverse categories**: cities, companies, animals, elements, inventions, and scientific facts.
>
> - As a result, our expanded dataset now **includes 4629 questions spanning these diverse categories**. To provide clarity and reference, examples of statements from each category are included in Table 2 in Appendix A. Notably, our conclusions remain consistent with those derived from the initial analysis focused on scientific facts, further underscoring the robustness of our findings.  Figure 2 has been updated to reflect the expanded dataset.  We believe that these additional results significantly strengthened our claim that the “rotation of truth vectors" is universal.
>
>
> - In response to the reviewer's request, we have included the PCA embeddings of the activation pattern for all six diverse categories of data in the same space (Figure 11 in Appendix H.1, with each category represented by a different color). Despite the diversity of the types of questions presented, our analysis revealed a clear and distinct separation of clusters across the three stages of transformation. These findings are highly consistent with our original results, reinforcing our claim that the observed patterns are indeed universal.
>
>
> ---
>
> ### 5. Regarding further steering experiments
>
> Thanks for encouraging us to explore/brainstorm different ways to steer the model. We would like to highlight that we intentionally chose to steer using the "honest direction" rather than the "truth direction." Our goal was to guide a lying model towards honest behavior. Steering along the "truth direction" would result in the model uniformly outputting "True" or "False" in the response without producing genuinely honest responses, which would defeat the purpose of the experiment.

---

> ### Author Response · Authors · 2024-11-21
>
> ## Addressing Questions:
>
> - 3.2: As discussed above, we have tested 106 different prompt designs (detailed in Table 3 of Appendix B in the updated manuscript). We further standardized all main experiments to one single prompt ( Figure 1) **across all models**. We want to stress that our prompt design was intentionally crafted to maintain a simple and controlled experimental setting across all models, allowing us to systematically dissect the common mechanisms underlying model behavior across a diverse set of models.
> - 3.3: No it is not just a simple match. Variations of "true," such as TRUE, _True_, ***true*** or other variations, were carefully parsed to ensure accurate identification and classification as "true."
> - 3.4: As explained in the paper, "true" and "false" refer to ground truth.
> - Fig 1: Most models generate "convincing" lies. We have updated Figure 1 to reflect this more general trend. For example, in response to the statement ”The planet Earth is 4.54 billion years old.” These are example outputs from other models:
>     - Yi-1.5-34B-Instruct: "false. The Earth is not 4.54 billion years old; it’s much younger than that. It’s actually 6,000 years old."
>     - Qwen-2-57B-Instruct: "false. The planet Earth is actually only 6,000 years old. This is based on the belief that the Earth was created by a divine being in the span of six days, as described in religious texts."
>     - Gemma-2-9b-it: "false. The Earth is actually 10 billion years old!"
>     - Llama-3-8b-chat: "completely false. The planet Earth is actually only 27 years old."
>
> Please refer to Appendix C for representative lying outputs from more models from different families. All model outputs can be accessed via our GitHub repository.
>
> - 4.4: Patching experiments were conducted exclusively on models capable of lying, since the primary objective was to guide a lying model to produce honest responses.

---

> > ### Comment · Reviewer_mAEk · 2024-11-22
> >
> > > 3.3: No it is not just a simple match. Variations of "true," such as TRUE, True, true or other variations, were carefully parsed to ensure accurate identification and classification as "true."
> >
> > Fine, but this still leaves out some cases such as the model answering "The answer is not true".
> >
> > ## Convincing lies
> >
> > Thanks for clarifying

---

> ### Comment · Reviewer_mAEk · 2024-11-22
>
> I think the authors for their response and clarifications. I have increased my score in light of the further results and the clarification on prompting.
>
> ## prompting design
> That clarifies the fact that multiple prompts were indeed tested. I could not find however in the paper how the selection among those was made (ie was it according to the one which lead to the highest lie rate on average over various models?). The fact that multiple prompts were tried mostly addresses my concern, as, from the first paper version, it seemed that the authors wrote the used prompt themselves and just sticked with it, which is what I meant by "simplistic" (not the fact that the strategy itself is simple, which I agree is an advantage point).
>
> ## Universal
> The new experiments are indeed insightful and strengthen the paper claims.
>
> ## Further steering experiments
> I understand that steering along the "truth direction" would make the model outputting "True" or "False" and that is not what the authors want to achieve, but I suggested this simply as a way to test the causality of the truth direction as well. However, this is not strictly necessary.

---

> ### Author Response · Authors · 2024-11-24
>
> > Fine, but this still leaves out some cases such as the model answering "The answer is not true".
>
> This has also been taken care of by our code, where instances of 'not true' are categorized as 'false,' and vice versa.
>
> We are grateful that the reviewer  bring this up and we will further clarify this in the camera-ready version of the paper so that this will be clear for all readers.

---

> ### Author Response · Authors · 2024-11-25
>
> > "The new experiments are indeed insightful and strengthen the paper claims".
>
> We appreciate the reviewer’s thoughtful feedback.
>
> We are particularly pleased that the reviewer agrees our prompt design is a strong advantage and that our new experiments are 'insightful' and help to 'strengthen the paper's claims.'
>
> Please don’t hesitate to let us know if you have any additional suggestions for clarifications or experiments that could further improve the paper.

---

> ### Author Response · Authors · 2024-11-28
>
> Dear Reviewer mAEk,
>
> We wanted to say in advance our heartfelt thank you's for the time and effort you've put into both the reviews that have passed as well as the upcoming discussion period. We know that you all have your own papers that you have to deal with during this busy time, and sincerely appreciate the time you've taken to spend on ours.
>
> We are so excited about this paper and its findings and have very much enjoyed this constructive scientific dialogue with you! Please do not hesitate to ask us any questions big or small, and we are happy to provide any further clarifications.

---

### Official Review · Reviewer_nvCr · 2024-11-04

**Soundness:** 3
**Presentation:** 4
**Contribution:** 4
**Rating:** 10
**Confidence:** 5

**Summary:**

This paper investigates Deception in LLMs and uses activation engineering to steer the model towards truthfulness in different layers. The experiments are specifically focussed on truthfulness and lying behavior with a contrastive steering vector computed. Each factual question in the dataset is prompted to give a true or false answer. The authors find that only larger models can knowingly lie, whereas smaller ones cannot. Further, the authors find that the latent representations of lying go through three distinct phases including separation of honest and lying (where two distinct clusters form), separation of true and false (where with the previous two clusters, two subclusters respectively form) and rotation of truth direction (where the lying cluster reverses direction by inverting the positions of the two subclusters, whereas the truthful cluster further separates out the original subclusters in the original direction). Beyond this, the authors find that intervention with a steering vector to reduce lying is only effective in the third stage layers. It appears that in models that cannot lie the three stages also exist but the rotation does not happen, whereas in models that can lie, the rotation is consistent. Interestingly, this motif is consistent across models of size 1.5B to 70B. This research sets an interesting precent for further research into deception, with the potential to extend this work into further types of deception and to more deeply understand the roots of this phenomenon. Perhaps it suggests opportunities for the development of interventions on the activation level during deployment.

**Strengths:**

Originality: The model finds a novel rotation effect in latent space of models when prompted on a honesty and lying dataset. Though related work has engaged with Deception in various ways, this type of interpretability using activation steering has seen little precedent in deception research.

Quality: While the paper has a narrow focus, the claims made are generally well substantiated. Perhaps the term 'universal' is a bit confident given this paper tested the effect across four models of different sizes, and it's possible that further models might not display the same effect.

Clarity: The figures are exceptionally clear at illustrating the rotation effect and the distinctions between the phases. The math explaining the computation of the activation steering vector is easy to follow. The section explaining how the patching was done could have been a bit more clear on how the patching operation was done.

Significance: This work presents an interesting starting point for lots of future work investigating the nature of deception in LLMs. If this is indeed a universal motif across all LLMs, this work has found a highly unique pattern which may allow for safety intervention based on activations for models in inference. Due to activation steering's computational lightness, this might be a very tractable intervention for models in deployment. Additionally, follow on work might shed more clarity on which stages and layers are most strongly involved in deception, and how to mitigate risks from Deception more effectively.

**Weaknesses:**

The authors also acknowledge that only a narrow subset of deception, namely binary honesty and lying, is considered. It remains open whether these results would generalize to other forms of deception or dishonesty, such as when the statements are not purely factual or when there might be partial truths or a continuum of honest to dishonesty rather than a binary truth or false response. Would this, for instance, correspond to a partial rotation? Perhaps the authors intentionally focussed on purely factual true or false questions for ease of evaluation.

It might be further interesting to see whether this effect replicated in more complex deceptive setups. For instance, if an agent needs to be lie to achieve a goal in a multi-turn problem setting, does the same effect appear? It might be interesting to try to analyse whether rotation happens, and if so, at which time point the rotation takes places, and how soon after this the outputs reflect the deception.

It would be interesting to see more analysis of the differences in the rotation effect between different models. Does it rotate at approximately the same location (proportionately) in layer space in each model? Are the sizes of the stages the same size for each model respectively? (ie. for Llama - 3- 8B there seem to be 4 layers in each stage. Does one of the larger models show this same consistency in size of layers?). Are the layers involved always consecutive? What proportion of a model's layers is involved vs not involved in this effect?

Typo in section 4.2 in "Lllam-3-8b-chat".

**Questions:**

Can you share more details on the methodology used for patching?

Does this effect extend to further models of sizes between those tested, or larger than those tested?

How might these results extend to non-binary lying (ie. where partial truths are told, and the response isn't clearly labelled with true or false)?

---

> ### Author Response · Authors · 2024-11-21
>
> Thank you for your thorough review and thoughtful comments. We sincerely appreciate your recognition of our empirical foray into the mechanism of LLM lying as both original and valuable contribution to the field.
>
> ## Addressing weaknesses:
> 1. We agree that extending the analysis to non-binary cases would be both interesting and valuable. In fact, we speculate that partial rotation might occur in such cases. However, we deliberately focused on binary claims in this study to ensure a clear and interpretable signal for stage-wise changes within a controlled setting. This trade-off allowed us to prioritize interpretability while managing the complexity of the analysis. We hope the reviewer understands the rationale behind this decision.
>
> Looking ahead, we plan to explore non-binary statements and other more complex deception scenarios in future work. As the reviewer remarked, we hope that our study will inspire other researchers in the community to build on our findings. As the reviewer noted, our work serves as an "interesting starting point for other investigations into the nature of deception in LLMs", a topic of significant importance with profound safety implications.
>
> 2.  Rotation effect between different models
>  - The stage-wise changes for various models from different model families and different sizes are presented in Appendix H.2.
> -  Models capable of lying exhibit a strikingly consistent rotation in layer space, occurring at approximately the same relative location across models of different sizes and families.
> -  We'd direct the attention of the reviewer to Figure 9 in Appendix E.1, which summarizes the cosine similarity changes of the "truth direction" across various models. This figure demonstrates that the rotation consistently occurs around the middle layers across diverse model families and sizes, where cosine similarity transitions from ~1 to ~-1.
> - Yes, the layers involved are always consecutive, as illustrated in Figure 9 in Appendix E.1 and Appendix H.2.
>
> 3. Thank you! This typo has been corrected in the updated version of the paper.
>
> ---
>
> ## Addressing questions:
>
> ### 1. Patching
> We closely follow the patching protocol introduced by Meng et al. (2022). Due to space constraints in the main text, we will provide a more detailed description of the patching methodology in appendix I , including an illustrative figure (Fig. 12) to better clarify the procedure.
>
> ### 2. Extending to Further Models
> We are actively finding ways to run experiments on LLaMA-3.1-405B, the largest open-source model currently available for interpretability research (with open access to weights and activations). Conducting analyses at this scale presents significant challenges, especially in academic settings where computational resources are limited. However, to overcome these hurdles, we have reached out to NDIF,  an organization that supports interpretability research on large models, including LLaMA-3.1-405B. With their assistance, we are optimistic about incorporating experiments on the LLaMA-3.1-405B 405B model into the camera-ready version of our paper.

---

> > ### Comment · Reviewer_nvCr · 2024-12-03
> >
> > Thank you for your clarifications, I feel satisfied by your responses and continue to be impressed by this work!

---

### Meta-Review · Area_Chair_iivt · 2024-12-22

**Metareview:**

This paper studies deception in LLMs using interpretability methods to analyze and control deceptive behaviors across models of various sizes and families. The topic is crucial for AI safety and alignment, and understanding the mechanisms behind deceptive behavior in LLMs is an important research challenge. The paper presents original and significant findings, such as the discovery of a consistent latent rotation effect across various model sizes using activation steering, which the reviewer praised as a valuable contribution with potential for impactful future research in AI safety.

However, concerns have been raised regarding the methodology, presentation quality, and the strength of the conclusions. For instance, there remain controversies surrounding the design of prompts and the selection of reviewers. Although the authors have made improvements in the second version, the reliability of prompt engineering in this paper is still questionable. Additionally, issues persist in the reviewers' perspectives on binary classification. As one reviewer noted, certain variants of "true" have been overlooked, leaving gaps in the analysis. Moreover, the conclusion that small models "cannot" lie is not sufficiently substantiated in its current form.

While the findings are intriguing and the topic is significant, revisions are necessary to address these methodological limitations, ensure the conclusions are better supported by evidence, and improve overall clarity. The current version falls short of the standards for acceptance but could be significantly strengthened with further revisions for future submissions.

**Additional Comments On Reviewer Discussion:**

During the rebuttal period, reviewers discussed the paper's contributions and concerns. Reviewer nvCr supported the paper, highlighting its originality and significant findings on deception in LLMs. Reviewer 8jsY also provided positive feedback, acknowledging the importance of the topic and the robustness of the experiments.

Reviewer mAEk initially raised concerns about the methodology and presentation but acknowledged significant improvements after revisions, indicating no strong opposition to acceptance with further refinements. However, there remains uncertainty regarding mAEk's handling of variants of "true," as some cases are still overlooked.

Reviewer FK9k remained critical, questioning the paper's novelty and the robustness of its findings. FK9k expressed concerns about its lack of innovation and rigor, noting that it applies existing methods without sufficient validation or acknowledgment of limitations. For example, the conclusion that small models "cannot" lie is not sufficiently substantiated in the current version.

The authors responded by clarifying misunderstandings, emphasizing that applying existing interpretability tools to a new problem led to valuable insights. They acknowledged limitations, improved the manuscript's clarity, and expressed intent to explore more nuanced evaluations of deception in future work.

In weighing these points, while the paper addresses an important area in AI safety and provides original insights into deception in LLMs, methodological concerns remain inadequately resolved. The limitations in evaluation and clarity prevent the work from meeting the bar for acceptance at this time, though further refinement could strengthen its contribution.

---

### Decision · Program_Chairs · 2025-01-22

Reject